# The Impact of Benevolent Sexism on Women’s Career Growth: A Moderated Serial Mediation Model

**DOI:** 10.3390/bs15010059

**Published:** 2025-01-12

**Authors:** Shuang Song, Po-Chien Chang

**Affiliations:** School of Business, Macau University of Science and Technology, Macau 999078, China; 3220000843@student.must.edu.mo

**Keywords:** benevolent sexism, self-esteem, emotional exhaustion, career growth, career development strategies, cognitive-affective personality system theory

## Abstract

This study investigates how benevolent sexism impedes women’s career growth, focusing on the mediating roles of self-esteem and emotional exhaustion and the moderating role of career development strategies. Using a three-wave, time-lagged survey, data from 410 female employees across various industries in China were analyzed with SPSS 24.0 and Mplus 8.3. Results indicate that benevolent sexism negatively influences career growth via reduced self-esteem and increased emotional exhaustion. Moreover, career development strategies mitigate this adverse effect, weakening the relationship between benevolent sexism and career growth. Higher levels of career development strategies lessen the detrimental impact of benevolent sexism on women’s professional progress. These findings enrich Cognitive-Affective Personality System theory by clarifying the mechanisms through which benevolent sexism undermines career development. They also highlight the practical significance of adopting robust career strategies to promote workplace gender equality and offer empirical insights into the broader implications of benevolent sexism on women’s career advancement.

## 1. Introduction

Despite significant progress in gender equality, women in China still encounter substantial obstacles in the workplace, particularly in attaining senior management positions and achieving salary parity. For instance, in 2023, women held only 33% of senior management roles, and their average monthly salary was approximately 87% of that earned by men ([67]). This is often referred to as the “glass ceiling”, highlighting the significant challenges women face in advancing in their careers or reaching senior leadership roles.

China’s socio-cultural landscape is profoundly shaped by Confucian traditions, which continue to exert significant influence on societal perceptions of gender roles. Central to Confucian philosophy is the concept of harmony within hierarchical structures, wherein women have historically been assigned roles centered around familial responsibilities. These traditional norms perpetuate gender stereotypes in the workplace, positioning women in supportive or subordinate roles and reinforcing the notion that they are better suited to “follow” rather than “lead” ([43]). This cultural backdrop, combined with a widespread suspicion of women’s leadership capabilities, creates significant barriers to women’s career advancement.

Women who display leadership traits are often perceived as less feminine, whereas those who exhibit relational qualities may be viewed as weak leaders ([44]). Such stereotypes adversely affect women’s opportunities in recruitment, task assignment, training, and performance evaluation. As society progresses, overt gender biases gradually diminish, but they still exist in more subtle and imperceptible forms ([44]).

Ambivalent Sexism Theory ([25], [27]) posits that sexism consists of two complementary components: hostile sexism (HS) and benevolent sexism (BS). HS portrays women as manipulative and inferior to men, while BS idealizes women as pure, moral, and deserving of men’s protection and support. In contrast to hostile sexism, which carries overtly negative connotations, BS is linked to positive stereotypes and favorable evaluations of women ([25], [26]). BS is more prevalent and socially accepted in contemporary society, with women being just as likely as men to embrace these attitudes ([26]; [5]). Despite its seemingly positive tone, BS actually reinforces traditional gender roles and legitimizes gender inequality. It portrays women as dependent on men, praising traditional female roles and qualities while also considering women fragile and in need of men’s protection ([18]). Though BS appears positive, it ultimately upholds traditional gender roles and supports gender inequality ([37]).

In particular, BS increases anxiety among women in the workplace ([59]) and leads them to emphasize their relational traits over task-related traits ([2]). Even when women hold high positions, those who demonstrate a leadership style that is harsh and decisive and deviates from traditional expectations often arouse the dissatisfaction of those with prejudices, which restricts women’s leadership styles to a limited range ([17]). BS, unlike overtly negative behaviors, is often wrapped in good intentions, making it harder to detect, and its effects and mechanisms harder to identify and quantify ([52]).

This study investigates the influence of BS in the workplace on women’s career advancement. Employing the framework of the Cognitive-Affective Personality System (CAPS) theory, this research will explore how BS manifests and its role in perpetuating the glass ceiling effect ([25]). Therefore, the goal of this study is to develop a deep theoretical understanding by assuming that the impact of BS on career growth is conditional and indirect, that is, that BS affects women’s career growth through specific mediating relationships. The CAPS theory posits that personality is comprised of cognitive-affective units (CAUs), which encompass the encoding and evaluation of the self, others, and situations, along with persistent goals, expectations, beliefs, abilities, and emotional states ([25]). The theory emphasizes that different situations activate different CAUs, leading to varied behavior patterns in the same individual across different contexts. However, BS does not directly affect career growth. Instead, it influences external work performance and behavior by impacting individuals’ cognitive and emotional systems. BS diminishes self-esteem, prompting negative self-assessments ([8]) and increasing emotional exhaustion, which makes individuals more susceptible to fatigue and helplessness under work pressure ([20]). This, in turn, negatively affects their work performance. Therefore, we infer that the relationship between BS and career growth may be serially mediated by self-esteem and emotional exhaustion (see Figure 1).

The CAPS theory posits that self-regulation plays a crucial role in shaping behavior and motivation ([53]). Empirical studies have found that self-regulation can improve individuals’ work engagement and career development ([38]; [48]). Hence, this study expects that career development strategies (CDS), functioning as a form of self-regulation, can alleviate the negative impact of BS by improving individuals’ professional skills and self-recognition. CDS can empower individuals to manage their careers more effectively. Consequentially, CDS are expected to moderate the relationship between benevolent sexism and career growth.

This study makes three contributions. First, this study utilizes the CAPS theory to analyze the mechanisms through which BS impacts career growth via cognitive and emotional dual pathways. By introducing self-esteem and emotional exhaustion as mediating variables, the research enriches the understanding of these processes. Second, this study enhances the understanding of BS. The findings provide organizational leaders with a deeper understanding of BS, enabling them to identify and implement strategies to mitigate its adverse effects. The insights derived from this research guide policymakers in crafting fairer workplace policies and practices. Third, by promoting gender equality, organizations improve overall efficiency and create a more inclusive work environment, ultimately benefiting both employees and the organization as a whole.

## 2. Literature Review and Hypotheses

### 2.1. Benevolent Sexism and Career Growth

Benevolent sexism (BS) refers to an ostensibly positive but condescending attitude toward women, portraying them as needing protection and inherently different from men, thus limiting their roles in society ([25]). Career growth refers to the advancement and progression in an individual’s professional life, which includes the development of skills, taking on increased responsibilities, achieving promotions, and attaining overall career satisfaction. It encompasses achieving personal career goals and experiencing upward mobility within one’s chosen profession ([9]). Career growth focuses on the development of an individual in a specific organization at a specific time ([72]). According to the CAPS theory, we hypothesize that when women perceive BS in the workplace, their CAUs are activated, leading to changes in cognitive evaluation and affective state, which, in turn, affects work performance ([53]).

First, BS reduces women’s career aspirations. BS encourages women to believe in concepts such as “men work outside and women work inside” and “it is better to marry a good man than to succeed in career”. Once women internalize these beliefs, it further strengthens their endorsement of BS ([8]), thus hindering them from pursuing career success. [30] ([30]) argued that the more women identify with BS, the more they tend to prioritize romantic partnerships and family responsibilities over professional goals. Women who endorse BS will lower their independence and competitive spirit, leading them to adopt conservative or passive behavior patterns at work and avoid risky or challenging tasks ([54]). This will reduce their labor force participation rate ([10]; [58]).

Second, BS limits women’s career choices and promotions. BS might appeal to individuals because it offers a rationale for gender inequality that portrays it as complementary and mutually beneficial, thus enhancing life satisfaction by increasing the belief in the fairness of the status quo ([12]). However, because BS is wrapped in good intentions, women often fail to recognize its potential negative effects. Individuals who endorse BS are more likely to support policies that promote equal employment for women, but this support is typically limited to promoting women’s employment in traditionally feminine positions, such as teaching, nursing, or administrative roles, rather than masculine positions, which are traditionally dominated by men and associated with leadership, assertiveness, and competitiveness ([33]). Although BS appears to promote gender equality, it limits women’s employment opportunities and hinders women’s career development by maintaining occupational gender segregation and restricting women’s advancement in male-dominated fields. Research has shown that social attitudes toward female leaders are generally less favorable than toward male leaders, resulting in women facing greater challenges in obtaining leadership positions ([17]). This bias can lead to women having fewer opportunities to attain leadership roles and encountering more difficulties in succeeding in these positions.

Third, women who suffer from BS receive less career guidance and fewer job opportunities. Based on stereotypical beliefs about gender capabilities, managers often assign tasks requiring innovation and strategic thinking to male employees, while tasks related to organizing activities or team building are assigned to female employees ([2]). Not only does this limit women’s opportunities for professional skill development, but it also affects their visibility and career growth within organizations. Research has found that male managers may give female subordinates more verbal praise while assigning them fewer challenging tasks ([7]; [42]). At the interpersonal level, BS from managers diminishes the career assistance women receive. On an intrapersonal level, when women endorse BS, their capacity to identify and pursue career support is weakened ([34]). BS not only limits women’s participation in traditionally male-dominated fields but also undermines their confidence and opportunities in career choices, career growth, and leadership development. Therefore, we propose the following:

**H1.** 
*Benevolent sexism is negatively related to career growth.*


### 2.2. Benevolent Sexism and Self-Esteem

Self-esteem refers to an individual’s overall evaluation of sense of self-worth and self-respect ([63]). This evaluation can be positive or negative, reflecting the individual’s emotional attitude towards self-image and self-satisfaction ([45]). Self-esteem can be formed through different pathways: persistent levels of self-esteem, self-esteem in a specific task or job, and expectations of others ([50]). In the long term, self-esteem generally exhibits stability, but, in the short term, it can be significantly influenced by situational factors and social feedback ([50]). For example, establishing close relationships with others can lead to a short-term boost in self-esteem ([11]). Conversely, being compared to others can lead to a temporary decrease in self-esteem ([69]). This change reflects the individual’s self-regulatory processes in the face of challenging information ([63]; [50]; [69]; [4]; [40]).

In short, such events can temporarily affect self-esteem. In this study, BS influences how women perceive themselves and are perceived by others ([22]). Moreover, BS as a situational factor can lead to short-term changes in self-evaluations ([63]; [50]; [69]; [4]; [40]).

CAPS theory posits that self-esteem, as an important cognitive-emotional unit, is significantly affected by external situations ([53]). BS adversely affects women’s self-evaluation, triggers negative self-cognition, and further limits their motivation and ability for career development. At the same time, BS increases the burden of emotional labor and makes it difficult for women to maintain high performance in a high-pressure work environment. Current research indicates that BS affects women’s recognition of their abilities ([16]) by reinforcing their dependence and incompetence, especially when sexist attitudes are communicated by powerful people like supervisors ([71]). For instance, female employees often receive feedback based on personality traits, while male employees receive feedback based on skills and results ([13]; [56]), which can undermine women’s confidence in their abilities and self-worth, allowing women to rationalize and accept their lower status, thereby damaging their self-esteem ([12]).

Additionally, because BS is often disguised as positive, it is difficult for victims to identify and resist. Witnessing or experiencing BS can lead to increased self-objectification and body shame in women ([25]; [66]). When women are praised for their appearance rather than their abilities, they may start to internalize the belief that their value is primarily based on their looks, leading to body shame and a reduction in self-esteem ([66]; [23]).

**H2.** 
*Benevolent sexism is negatively related to self-esteem.*


### 2.3. The Mediating Effect of Self-Esteem

CAPS theory posits that situational factors first trigger individuals’ cognitive-affective units, which then influence behavior ([53]). Research has shown that women with lower self-esteem are less likely to pursue leadership positions, negotiate for higher salaries, or seek career advancement opportunities ([57]; [3]), particularly in traditionally male-dominated fields ([8]). This diminished self-esteem makes them more vulnerable and creates difficulty in aligning their personal identity with their professional roles ([68]), which negatively impacts job satisfaction and career growth ([53]). Hence, we believe that BS adversely affects career growth by reducing self-esteem, which, in turn, undermines career aspirations and motivation.

**H3.** Self-*esteem mediates the relationship between benevolent sexism and career growth.*

### 2.4. Benevolent Sexism and Emotional Exhaustion

Emotional exhaustion refers to a state of being emotionally and physically drained due to prolonged workplace demands. It is characterized by persistent fatigue, a lack of motivation or enthusiasm for work, and diminished effectiveness in task performance ([51]). Emotional exhaustion is defined as an extreme emotional and chronic workplace stress response that renders employees psychologically unable to perform their duties ([24]). According to the CAPS theory, specific cues or stimuli in different situations trigger individuals’ emotional and behavioral responses ([53]). Emotional exhaustion, as an affective manifestation of CAUs, develops through the emotional experience pathway. Perceived BS can undermine women’s sense of competence and trigger stress responses ([46]). This can lead to anxiety and self-doubt, which may transform into stable and specific individual behaviors, such as avoiding leadership roles, hesitating to voice opinions, or conforming to traditional gender expectations due to fear of judgment ([53]).

On the one hand, BS increases women’s anxiety and uncertainty, leading to a lack of confidence in their professional abilities ([20]), feelings of insecurity, and self-doubt ([16]). On the other hand, BS requires women to show certain emotions such as obedience and tenderness, which may conflict with the actual requirements of professional roles and increase the burden of emotional labor ([36]), thereby bringing additional pressure and an emotional burden to women ([19]).

Research indicates that workplace stressors, including task demands and emotional labor, are critical antecedents of emotional exhaustion ([51]). Discriminatory practices, such as the underestimation of women’s competencies in domains like mathematics and science or the presumption that specific occupations are more appropriate for men, exacerbate these stressors ([60]). BS exacerbates this dynamic by imposing additional emotional demands on female employees. To conform to societal expectations or workplace norms, women often expend greater self-regulatory resources and engage in emotional labor. This not only reinforces traditional gender stereotypes but also creates role conflict and amplifies emotional strain ([35]; [29]). As a result, BS is recognized as a threat to emotional resources. Therefore, we propose the following:

**H4.** 
*Benevolent sexism is positively related to emotional exhaustion.*


### 2.5. The Mediating Effect of Emotional Exhaustion

The CAPS theory suggests that individual behavior is not fixed but varies with the context, with emotional responses being activated to drive behavior ([53]). As we previously inferred, the ambiguous and often covert nature of BS requires victims to navigate and interpret its intentions, creating emotional strain ([1]). When female employees perceive BS, they often need to suppress their genuine emotional responses to meet workplace expectations. This prolonged suppression, referred to as “overextending self-control”, depletes their emotional resources ([65]). Simultaneously, efforts to protect their self-worth can lead to an “emotionally defensive” state, wherein they prioritize coping strategies over proactive engagement at work. This defensive state reduces their willingness and ability to invest effort into workplace tasks, ultimately lowering work performance and hindering career development ([39]). Our study suggests that emotional exhaustion is a mediating factor between BS and career growth. Therefore, we propose the following:

**H5.** 
*Emotional exhaustion mediates the relationship between benevolent sexism and career growth.*


### 2.6. The Chain Mediating Effect of Self-Esteem and Emotional Exhaustion

Individual behavior is influenced by the dynamic interaction of cognitive-affective units (CAUs), which are activated in response to specific situational features ([53]). In the context of BS, these CAUs may trigger a cascade of cognitive-affective responses that influence behavior. Specifically, BS can undermine self-esteem, a critical factor in determining how individuals perceive and respond to external circumstances. Individuals with low self-esteem are more susceptible to emotional distress and exhaustion, particularly when faced with negative outcomes that activate specific CAUs ([25]; [63]). Therefore, we propose the following:

**H6.** 
*Self-esteem and emotional exhaustion serially mediate the relationship between benevolent sexism and career growth.*


### 2.7. The Moderating Effect of Career Development Strategies

CAPS theory posits that self-regulation plays a vital role in behaviors and motivation, affecting individuals’ cognitive and emotional responses to situations ([53]). Self-regulation can be defined as the uniquely human ability to consciously control and adjust one’s thoughts, emotions, impulses, and behaviors to align with social expectations or personal goals ([53]). Empirical studies found that self-regulation can improve individuals’ work engagement ([38]; [48]). This study argues that career development strategies (CDS) have a self-regulation role, which can help individuals navigate their careers more effectively. CDS are classified into seven categories: creating opportunities, extended work involvement, self-nomination/self-presentation, seeking career guidance, networking, opinion conformity, and other enhancement ([28]). Individuals can evaluate the situation and choose adaptive ways to cope based on their experience, beliefs, ability levels ([21]), and sense of career control ([61]). Research showed that individuals can enhance self-efficacy in work through implementing career development strategies, such as skill improvement, network building, self-nomination ([47]). Consistently, we expect that an individual’s CDS can moderate the serial mediated relationships between BS and career growth via self-esteem and emotional exhaustion.

From a cognitive perspective, people derive their overall sense of self-esteem through how they feel about themselves in different social situations ([32]). Specifically, BS from the external environment may cause a short-term impairment to one’s self-esteem, whereas individuals can utilize subjective cognition and self-nomination strategies to mitigate the damage caused by BS, and enhance self-recognition and self-validation, thus bolstering their self-esteem in the long run ([40]). Moreover, work involvement is an important aspect of career development strategy ([28]). It refers to an individual investing energy and passion in work and having a deep connection with their work role. Employees who acquire achievement and recognition from work will have boosted job satisfaction and self-esteem, which, in turn, promotes greater work involvement ([38]; [48]).

From an emotional perspective, self-regulation strategies such as seeking guidance or building a social support network can alleviate stress and contribute to emotional resilience ([70]). Social support, especially support from supervisors, can serve as a job resource that alleviates the psychological and physiological stress caused by jobs, thereby enhancing individuals’ emotional stability ([49]). [28] ([28]) found that job type (manager vs. non-manager), career mobility (stable vs. unstable), and gender were all associated with the tendency to use specific career strategies. Research found that women often use seeking guidance and mentoring and expanding personal networks ([28]). CDS can buffer the negative impact of BS on self-esteem and emotional exhaustion by empowering individuals to counteract discriminatory behaviors and attitudes ([21]). Therefore, we propose the following:

**H7a.** 
*Career development strategies moderate the negative impact of benevolent sexism on self-esteem.*


**H7b.** *Career development strategies moderate the strength of the mediated relationship between benevolent sexism and career growth* via *self-esteem and emotional exhaustion, such that the serial mediation will be weaker in the high-level career development strategies than in the low-level career development strategies.*

## 3. Methodology

### 3.1. Study Design

This study employed snowball sampling to collect data from female employees across various industries in mainland China, aiming to achieve a diverse sample that represents working women in urban settings. From October 2023 to December 2023, we leveraged the professional networks of Doctor of Business Administration (DBA) students from the business school to contact HR managers from 18 companies in mainland China, spanning diverse sectors (e.g., accommodation and catering, wholesale and retail, education and training, medical and health, etc.). The goal was to include participants from various industries, career stages, and demographic backgrounds to enhance the generalizability of the findings. After thoroughly explaining the research objectives and obtaining informed consent, questionnaires were distributed to these organizations in collaboration with the HR managers.

To test the hypotheses and assess relationships among variables, this study utilized SPSS version 24.0 and the PROCESS macro (v4.1) by [31] ([31]) for descriptive statistics and hypothesis testing, including bootstrapped mediation and moderation analyses. Additionally, confirmatory factor analysis (CFA) was conducted using Mplus version 8.3 to evaluate the measurement model and validate construct validity.

To minimize common method bias (CMB)—a measurement error that occurs when both predictor and criterion variables are collected from the same source at the same time—data collection was conducted in three stages over a three-month period, with one-month intervals between each stage ([62]). Questionnaires featured varied response formats and scales to further mitigate CMB risks.

A power analysis was conducted using the PROCESS macro (v4.1) to ensure sufficient statistical power for the moderated serial mediation analysis. Assuming a medium effect size (f^2^ = 0.15), a significance level of 0.05, and a power of 0.80, the minimum required sample size was calculated to be 85 participants. Given the complexity of the study design, which involves serial mediation and moderation effects, a larger sample size was targeted to enhance the robustness of the findings. Consequently, 410 valid responses were collected, far exceeding the minimum requirement and ensuring the reliability and validity of the results.

### 3.2. Participants

In the survey collection stage, during the first stage (T1), respondents provided information on their understanding of benevolent sexism, along with their demographic information. In the second stage (T2), data on self-esteem and emotional exhaustion were collected. The third stage (T3) focused on gathering information on career development strategies and career growth. A total of 488 questionnaires were collected, of which 410 were deemed valid after excluding male respondents and incomplete responses, yielding an effective response rate of 81.03%.

Among the 410 valid questionnaires, the largest age group was 31–40 years old (22.44%), followed by 18–25 years old (22.20%) and 41–50 years old (20.49%), aligning with the study’s focus on career development. In terms of education, 48.78% (SD = 1.25) of respondents held at least a bachelor’s degree. Participants worked in various industries, notably accommodation and food services (32.44%), wholesale (18.05%), and education/training (10.73%), providing a broad view on women’s careers. Most respondents were ordinary employees (48.05%) (SD = 0.83), followed by junior managers (32.68%) and middle managers (16.34%), suggesting that the majority of participants held either general staff or junior management positions.

### 3.3. Variable Measurement

All scales used in this study have been extensively employed in empirical research and have undergone rigorous validation and reliability testing. All instruments were translated into Chinese using the back-translation method to ensure both linguistic and conceptual equivalence. A bilingual expert panel reviewed the translations to confirm cultural appropriateness and semantic clarity. Each scale utilized a 5-point Likert scale, with responses ranging from 1 to 5 (1 = strongly disagree, 5 = strongly agree).

Benevolent Sexism: Benevolent sexism was assessed using the 19-item Benevolent Sexism in the Workplace Scale developed by [71] ([71]). Participants responded to statements about their experiences of benevolent sexism in the workplace. A sample item is “In your organization, men typically take leadership roles because they are perceived as better equipped to handle difficult decisions”. The Cronbach’s alpha coefficient for this scale was 0.94.

Self-Esteem: Self-esteem was measured using the 10-item Self-Esteem Scale developed by [63] ([63]). Participants rated statements about their self-worth and confidence. A sample item is “I am satisfied with myself”. The Cronbach’s alpha coefficient for this scale was 0.876.

Emotional Exhaustion: Emotional exhaustion was assessed using the 9-item Emotional Exhaustion Scale of the Maslach Burnout Inventory ([51]). Participants indicated their levels of emotional exhaustion related to their work. A sample item is “I feel emotionally drained from my work”. The Cronbach’s alpha coefficient for this scale was 0.76.

Career Development Strategies: Career development strategies were assessed using the 26-item Career Strategies Inventory developed by [28] ([28]). Participants responded to statements regarding the strategies they use to enhance their career development. A sample item is “I frequently seek feedback on my performance”. The Cronbach’s alpha coefficient for this scale was 0.91.

Career Growth: Career growth was measured using the 15-item Career Growth Participants rated statements about how their current job has contributed to their career progression. The scale was developed by Weng and Xi ([72]). A sample item is “My current job has brought me closer to my career goals”. The Cronbach’s alpha coefficient for this scale was 0.93.

Control Variables: To reduce potential confounding effects, we included age, education, current position, and industry as control variables in our analysis.

## 4. Results

### 4.1. Confirmatory Factor Analysis

We assessed the discriminant validity of all variables in our study by conducting a series of confirmatory factor analyses (CFAs) using Mplus 8.3 ([55]). As shown in Table 1, the hypothesized five-factor model (χ2 = 2713.51, df = 1700, TLI = 0.90, CFI = 0.91, RMSEA = 0.04, SRMR = 0.05) demonstrated the best fit compared to the alternative models. Furthermore, the hypothesized model was compared with all alternative models using chi-square difference tests, which indicated that our hypothesized model provided the best fit for the data, thus supporting the discriminant validity of the measurement ([64]).

### 4.2. Descriptive Statistics and Correlations

The descriptive statistics and correlations presented in Table 2 reveal a significant negative correlation between X (BS) and Y (career growth) (r = −0.47, *p* < 0.01). This suggests that higher levels of BS are associated with reduced career growth among females. Additionally, a significant negative relationship was observed between X (BS) and M1 (self-esteem) (r = −0.24, *p* < 0.01), indicating that increased exposure to BS is associated with lower self-esteem. Furthermore, X (BS) was positively correlated with M2 (emotional exhaustion) (r = 0.21, *p* < 0.01), suggesting that individuals who experience higher levels of BS also tend to experience greater emotional exhaustion. The negative correlation between M2 (emotional exhaustion) and Y (career growth) (r = −0.24, *p* < 0.01) further suggests that higher levels of emotional exhaustion are associated with more significant challenges in career advancement. These findings provide preliminary support for the hypothesized relationships among the variables and satisfy the necessary conditions for testing serial mediation.

### 4.3. Hypothesis Testing

We used SPSS to conduct the data analysis, utilizing the PROCESS macro (v4.1) for moderated serial mediation analysis ([31]). The analysis focused on evaluating the direct and indirect effects of X (BS) on Y (career growth), mediated through M1 (self-esteem) and M2 (emotional exhaustion). Additionally, we investigated the moderating effect of career development strategy on these proposed relationships. The results of these analyses are detailed below for each hypothesis.

Table 3 presents the estimated regression coefficients, revealing several significant relationships among the variables. The analysis indicates a significant negative association between X (BS) and Y (career growth) (b = −0.419, *p* < 0.001), thereby supporting H1. Additionally, X (BS) was found to have a significant negative relationship with M1 (self-esteem) (b = −0.300, *p* < 0.001), which supports H2. H3 posits that M1 (self-esteem) mediates the relationship between X (BS) and Y (career growth). This hypothesis is supported by a significant indirect effect of X (BS) on Y (career growth) through M1 (self-esteem) (effect = −0.035, SE = 0.014, 95% CI [−0.066, −0.009]), as detailed in Table 4. Furthermore, H4 is also supported, with a significant positive relationship identified between X (BS) and M2 (emotional exhaustion) (b = 0.174, *p* < 0.001). Finally, H5 proposes that M2 (emotional exhaustion) mediates the relationship between X (BS) and Y (career growth), which is corroborated by a significant indirect effect of X (BS) on Y (career growth) through M2 (emotional exhaustion) (effect = −0.025, SE = 0.012, 95% CI [−0.052, −0.004]).

To test Hypothesis 6, we evaluated both the specific and total indirect effects of X (BS) on Y (career growth), considering pathways through at least one mediator and through the two serial mediators. Serial mediation effects were analyzed using the bootstrap method with 5000 resamples. As presented in Table 4, the indirect effect of X (BS) on Y (career growth) via M1 (self-esteem), followed by M2 (emotional exhaustion), was significant (effect = −0.006, SE = 0.004, 95% CI [−0.015, −0.001]). These findings provide support for Hypothesis 6.

### 4.4. Moderation Effect Testing

To test H7, which posits the moderating effect of career development strategy on the relationship between X (BS) and M1 (self-esteem), we first analyzed the moderation effects. Table 3 summarizes the moderation effect analysis for self-esteem. The significant interaction term (BS × career development strategy, β = 0.36, *p* < 0.001) indicates that the impact of BS on self-esteem varies based on the level of career development strategy. Specifically, the negative effect of BS on self-esteem is weaker for individuals with a stronger career development strategy. Covariates such as age, level, and education did not significantly influence the outcome.

To illustrate the interaction effects, regression lines were plotted for career development strategy values at the mean and ±1 standard deviation (Figure 2), maintaining it as a continuous variable. The negative impact of BS on self-esteem was significant only for individuals with low levels of career development strategy (−0.306, 95% CI = [−0.447, −0.166]) and non-significant for those with high levels (0.060, 95% CI = [−0.084, 0.203]). These findings support H7a.

In addition, Table 5 evaluates the moderated serial mediation effect. At high levels of career development strategy, the indirect effect of BS on career growth via self-esteem and emotional exhaustion is 0.001 (95% CI = [−0.002, 0.005]), showing non-significance. However, at low levels of career development strategy, the indirect effect is −0.006 (95% CI = [−0.015, −0.001]), demonstrating a significant negative impact. These results confirm that lower levels of career development strategy amplify the negative effects of BS on career growth through self-esteem and emotional exhaustion, thereby supporting H7b.

## 5. Discussion

Drawing on Cognitive-Affective Personality System theory, this study examines the relationship between benevolent sexism (BS) and career growth, focusing on the mediating roles of self-esteem and emotional exhaustion, as well as the moderating role of CDS. The findings reveal a negative relationship between BS and career growth. Furthermore, CDS moderate the detrimental effects of BS on self-esteem, influencing the serial mediation effects of self-esteem and emotional exhaustion on career growth. Employees with lower levels of CDS are more vulnerable to the negative impacts of BS, while enhancing CDS can buffer these adverse effects. The results validate Hypotheses 1 through 5, demonstrating that self-esteem and emotional exhaustion, as core CAUs, significantly explain the pathways through which BS impacts career growth. Hypotheses 6 and 7 further establish that CDS moderate the negative effects of CAUs, thereby providing additional support for career development. By identifying CAUs as critical mechanisms within the CAPS framework, this study deepens our understanding of how BS influences workplace outcomes and highlights the protective role of CDS. Our findings provide valuable insights into the theoretical framework and practical applications, which are elaborated below.

### 5.1. Theoretical Implication

These findings enrich the Cognitive-Affective Personality System (CAPS) theoretical framework in three key ways.

First, this study empirically validates the CAPS theory’s proposition that situational factors, such as perceived benevolent sexism, activate cognitive-affective units that influence individual behavior. The activation of cognitive-affective units explains how benevolent sexism undermines career growth through diminished self-esteem and increased emotional exhaustion. Specifically, benevolent sexism acts as a situational trigger that alters self-perceptions and emotional states, ultimately constraining career aspirations and job performance. These results provide strong evidence for CAPS theory’s emphasis on the interaction between cognitive and emotional systems in shaping behavior ([25]; [15]).

Second, by identifying self-esteem and emotional exhaustion as serial mediators, this study contributes to a deeper understanding of the internal mechanisms through which benevolent sexism affects career growth. Self-esteem’s role in mediating the cognitive pathway, and emotional exhaustion’s role in the affective pathway, illustrate how cognitive-affective units interact dynamically under conditions of benevolent sexism. This dual-pathway perspective advances CAPS theory by demonstrating the interplay between cognitive and affective responses in workplace contexts ([6]).

Third, we introduce career development strategies (CDS) as a boundary condition that moderates the mechanism through which benevolent sexism influences career growth. Given that self-esteem is a key resource for enhancing self-efficacy and achieving career growth ([41]), we reveal the effectiveness of CDS in enhancing women’s self-esteem and promoting career growth. This finding aligns with CAPS theory’s emphasis on self-regulation as a crucial factor in adaptive behavior. By demonstrating how CDS moderate the relationship between benevolent sexism and career growth, this study extends CAPS theory’s applicability to organizational and career development contexts ([53]; [28]).

### 5.2. Practical Implications

This study offers three main practical implications.

First, our findings confirm that benevolent sexism negatively impacts women’s career growth by reducing self-esteem and increasing emotional exhaustion. Organizations should raise awareness of gender bias through comprehensive training for HR professionals and leadership teams. For instance, workshops and educational programs can help employees and managers recognize and mitigate the subtle and harmful effects of benevolent sexism ([25]; [71]).

Second, recruitment, evaluation, and promotion processes must be revised to eliminate gender bias. Implementing fair, transparent, and standardized criteria will minimize subjective judgments influenced by gender stereotypes. Additionally, providing psychological support and career development counseling can empower women to enhance their self-efficacy and better manage emotional exhaustion ([49]). A supportive organizational culture that explicitly promotes diversity and inclusion through policies aimed at combating gender bias is essential to counteract these systemic issues ([36]).

Third, enhancing management’s awareness of gender equality through targeted seminars and training courses is critical. Managers need to develop the skills to identify and address gender bias effectively. Moreover, strengthening CDS, such as offering leadership training, networking opportunities, and mentorship programs, can help women plan their career paths and access growth opportunities ([28]). These measures not only support women’s professional advancement but also contribute to improved organizational performance and competitiveness.

### 5.3. Research Limitations and Prospects

First, convenience sampling, while practical and efficient for accessing respondents across diverse industries, may not fully capture the broader population of working women in China ([14]). To enhance the representativeness of future research, we recommend employing probability sampling methods, such as stratified or cluster sampling, to ensure that the sample more accurately reflects the diversity of working women in China. Stratified sampling in particular could be used to capture women across different industries, educational levels, and geographic regions, thereby reducing sampling bias and improving generalizability.

Second, reliance on self-reported data introduces biases such as social desirability effects, which can affect data reliability ([62]). To address this, future studies should incorporate multiple data sources, such as peer reports or objective performance measures, to cross-validate the findings. Additionally, expanding the sample size could further enhance the external validity of the results.

Third, the cross-sectional design of this study only reveals correlations and cannot establish causality ([17]). Future research should employ longitudinal or experimental study designs to track long-term changes in career growth and explore possible predictors.

Finally, future research could introduce more variables, such as organizational culture, leadership styles, work environment, employee satisfaction, and mental health, to comprehensively explore the factors influencing the relationships between benevolent sexism and career growth. Expanding research to include male employees would also help researchers to explore the differential impacts of gender bias across genders, enhancing our understanding of gender bias mechanisms in the workplace ([42]).

## 6. Conclusions

This study advances our understanding of the Cognitive-Affective Personality System theory by elucidating how benevolent sexism impacts career growth through both self-esteem and emotional exhaustion. Specifically, our findings reveal that career development strategies serve as a crucial moderating factor, suggesting that the interplay between self-esteem, emotional exhaustion, and external career development strategies significantly shapes career progression for women in the workplace. These insights carry profound implications for gender equity policies. By shedding light on the impact of benevolent sexism, this study provides an empirical foundation for future interventions to promote inclusive and equitable career pathways.

## Figures and Tables

**Figure 1 behavsci-15-00059-f001:**
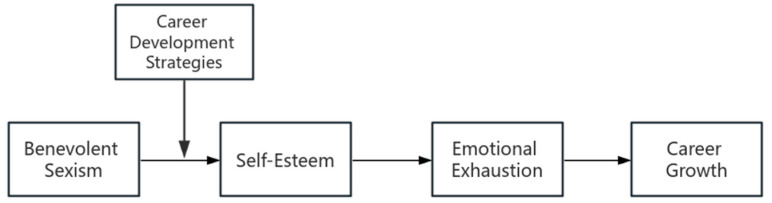
Proposed moderated serial mediation model.

**Figure 2 behavsci-15-00059-f002:**
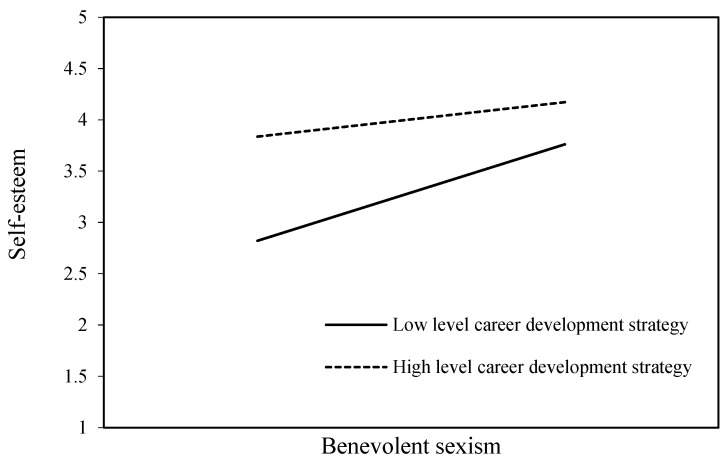
Moderating effect of career development strategy on the relationship between benevolent sexism and self-esteem.

**Table 1 behavsci-15-00059-t001:** The results of confirmatory factor analysis.

Measurement Model	χ^2^	df	∆χ^2^/(∆df)	TLI	CFI	RMSEA	SRMR
Five-Factor Model (BS; SE; EE; CG; CDS)	2713.51	1700		0.90	0.91	0.04	0.05
Four-Factor Model (BS; SE; EE; CG + CDS)	3371.28	1704	164.44	0.84	0.84	0.05	0.06
Three-Factor Model (BS; SE; EE + CG + CDS)	3821.17	1707	149.97	0.80	0.8	0.06	0.07
Two-Factor Model (BS; SE + EE + CG + CDS)	5042.64	1709	610.73	0.68	0.69	0.07	0.09
One-Factor Model (BS + SE + EE + CG + CDS)	6790.79	1710	1748.15	0.51	0.53	0.09	0.10

Note: N = 410; BS = benevolent sexism; SE = self-esteem; EE = emotional exhaustion; CG = career growth; CDS = career development strategy. Abbreviations: TLI = Tucker–Lewis index; CFI = comparative fit index; RMSEA = root mean square of approximation; SRMR = standardized root mean square residual.

**Table 2 behavsci-15-00059-t002:** Descriptive statistics and correlation coefficients of variables.

Variables	M	SD	1	2	3	4	5	6	7	8
1. Age	34.97	11.95								
2. Education level	3.31	1.25	−0.04							
3. Current position level	1.74	0.83	0.29 ***	0.12 *						
4. Benevolent sexism	2.41	0.69	−0.18 ***	0.23 ***	−0.01	(0.94)				
5. Self-esteem	3.78	0.82	0.15 **	0.07	0.16 **	−0.24 ***	(0.88)			
6. Emotional exhaustion	2.63	0.64	0.08	0.05	0.01	0.21 **	−0.21 ***	(0.76)		
7. Career growth	3.40	0.72	0.14 **	−0.08	0.07	−0.47 ***	0.27 ***	−0.24 ***	(0.93)	
8. Career development strategy	3.33	0.51	0.18 ***	−0.04	0.09	−0.48 ***	0.36 ***	−0.35 ***	0.53 ***	(0.91)

Note: N = 410; * *p* < 0.05, ** *p* < 0.01, *** *p* < 0.001. The values in parentheses along the diagonal are the reliability coefficients of the scales.

**Table 3 behavsci-15-00059-t003:** Regression results for moderated serial mediation model.

Predictor	M1: Self-Esteem				M2: Emotional Exhaustion		Y: Career Growth	
	Model 1		Model 2		Model 3		Model 4	
	B	SE	B	SE	B	SE	B	SE
Constant	3.88 ***	0.21	3.4 ***	0.16	2.48 ***	0.23	4.18 ***	0.26
Age	0.01	0	0	0	0.01	0	0	0
Level	0.12 *	0.05	0.08	0.05	−0.01	0.04	0.03	0.04
Education	0.07 *	0.03	0.05	0.03	0.01	0.03	0	0.03
X: Benevolent sexism	−0.3 ***	0.06	−0.12 *	0.06	0.17 ***	0.05	−0.42 ***	0.05
M1: Self-esteem					−0.15 ***	0.04	0.12 ***	0.04
M2: Emotional exhaustion							−0.14 **	0.05
W: Career development strategy			0.58 ***	0.09				
Benevolent sexism × career development strategy			0.36 ***	0.07				
R^2^	0.1		0.21		0.09		0.26	

Note: N = 410; * *p* < 0.05, ** *p* < 0.01, *** *p* < 0.001. Unstandardized regression coefficients are reported.

**Table 4 behavsci-15-00059-t004:** Serial mediation model.

Path	Effect	SE	Bootstrapping 95% CI	
			LLCI	ULCI
Total effect (benevolent sexism → career growth)	−0.48	0.05	−0.58	−0.39
Total indirect effect	−0.07	0.02	−0.11	−0.03
Specific indirect effect breakdown				
Benevolent sexism → self-esteem → career growth	−0.04	0.01	−0.07	−0.01
Benevolent sexism → emotional exhaustion→career growth	−0.02	0.01	−0.05	−0.00
Benevolent sexism → self-esteem → emotional exhaustion → career growth	−0.01	0.00	−0.02	−0.00

Note: X = benevolent sexism, M1 = self-esteem, M2 = emotional exhaustion, Y = career growth. Abbreviations: LL, lower limit; UL, upper limit; CI, confidence interval.

**Table 5 behavsci-15-00059-t005:** Moderated serial mediation model.

Moderating Variable	Path: Benevolent Sexism → Self-Esteem			Path: Benevolent Sexism → Self-Esteem → Emotional Exhaustion → Career Growth		
	Effect	Bootstrapping 95% CI		Effect	Bootstrapping 95% CI	
		LLCI	ULCI		LLCI	ULCI
High-Level Career Development Strategy	0.060	−0.084	0.203	0.001	−0.002	0.005
Low-Level Career Development Strategy	−0.306	−0.447	−0.166	−0.006	−0.015	−0.001

Note: N = 410. Bootstrap sample size = 5000.

## Data Availability

The datasets generated and/or analyzed during the current study are not publicly available due to ongoing research and analysis. However, they can be obtained from the corresponding author upon reasonable request.

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
