# Peer review of "The Impact of Benevolent Sexism on Women’s Career Growth: A Moderated Serial Mediation Model"

_behavsci, 2025, doi:10.3390/bs15010059_

Round 1

Reviewer 1 Report

Comments and Suggestions for Authors

Summary: This study aimed to assess the relationship between benevolent sexism and career growth in women and whether self-esteem and emotional exhaustion mediated the effect, and/or whether taking part in career development strategies moderated the effect. The manuscript is well organized and I appreciate the step-by-step nature of how the hypotheses and results are presented. My main concern (detailed in the major comments) is what is being measured with the benevolent sexism measure and what that means for the interpretation of the results. See the major comments for more specific detail. The minor comments identify where editorial changes are necessary.

Major Comments:

1.       The title could be improved by making more specific to the topic: The Impact of Benevolent Sexism on Women’s Career Growth: A Moderated Serial Mediation Model

2.       Line 33: “Rebound effect” and “double bias” are not known terms and should be defined/explained if used.

3.       Line 54: “Traditional sexism” should be defined and contrasted with HS and BS.

4.       Line 55: “Incivility” is not a known term and should be defined.

5.       Line 113: What is meant by “intimate relationships” in this context?

6.       Line 124: What is meant by “masculine positions”?

7.       Line 163: It’s unclear how one’s cognitive understanding of a situation is self-esteem. Is it more accurate to say that one’s cognitive appraisal of the situation may impact self-esteem?

8.       Line 199-200: The meaning of this sentence isn’t clear to me.

9.       Line 202: What are examples of “stable and specific individual behaviors” that would result from anxiety and self-doubt in response to BS?

10.   Line 215-215: It’s unclear how BS is a threat to resources.

11.   Line 222: It’s unclear what is meant by “overextend their self-control” or “emotionally defensive

12.   Line 243: Self-regulation needs to be defined in the manuscript

13.   Line 246-248: This sentence is confusing and needs grammatical improvements. Also, what is meant by visibility?

14.   Line 260: I think prejudiced is being misused in this sentence. I think the authors mean to say that something feeling like there is a prejudiced towards them may cause short term impairment to self-esteem. As it reads now, it’s as if the individual has a prejudiced attitude themselves. This notion applies to the entire manuscript, as detailed in the next comment.

15.   Throughout the Literature Review and Hypotheses section, I think it’s important to distinguish when the evidence presented about BS is referring to BS that is perceived by the individual woman and perpetrated by someone else compared to the effect of that individual woman’s own BS beliefs or actions. 

16.   Line 275: This sentence has hints of BS baked in (that women are in greater need for emotional support). Was that intended?

17.   Line 293: What is meant by “representative” sample? What population were you aiming to represent?

18.   Line 300: What is “common method bias?”

19.   It would be helpful to specify where the study took place (China? Specific areas of China?-this is in the abstract but not manuscript), given there may be cultural differences in workplace behaviors, gender roles, industries, etc. This could also be noted as a limitation/future direction (expand to other areas of the world where gender roles differ)

20.   There should be a section in the Methods describing the statistical/data analysis plan. This information should not be found solely in the results.

21.   I’m worried that the Benevolent Sexism scale used does not differentiate between one’s experiences of being the subject of BS versus their own sexist personal beliefs. The example item suggests that the scale assesses the latter, i.e., if I hold sexist beliefs, as opposed to “have other people’s sexists beliefs been thrust upon me. This is important because it changes the interpretation of the findings. For instance, the negative correlation with self-esteem would suggest that if I hold sexist beliefs, I’m more likely to have poor self-esteem versus, if I experience others having sexist beliefs, then I’m more likely to have poor self-esteem. This is a major difference.

22.   Line 363: Clarify what type of correlation testing was performed. (Spearman’s due to the ordinal nature of the data?)

23.   Suggest citing how SPSS was used to do this analysis and cite appropriate. I assume it was the PROCESS macro by Hayes?

24.   Line 299-404: It would be helpful to have the variable names, at least in parentheses) for X, Y, M1 and M2 to make this paragraph more understandable without having to refer to the table.

25.   Line 420: If I’m understanding the result correctly, this paragraph could be said more concisely and clearly. Something like: The negative relationship between BS and self-esteem was only significant for those with low levels of career development strategy (insert values) and was not significant for high career development strategy levels.

26.   How was career development strategy dichotomized for the analysis? Was a cutoff score used?

27.   How were each of the scales scored? Sum? Average? What was the range of scores possible?

28.   Was there a sample size or power analysis performed? How big of a sample was needed for this complex analysis?

29.   Line 446: I would be careful about saying the enhancing career development strategies promotes career growth, since the analysis showed that in those with higher career development strategy scores, there was no significant relationship (neither positive nor negative) between BS and career growth (unless I misunderstood the results).

Comments on the Quality of English Language

Minor Comments:

1.       Line 24-26: The first sentence of the introduction requires grammatical improvements.

2.       Line 49: Sentence is missing a word (“it ultimately upholds”)

3.       Line 56-57: Sentence needs grammatical improvements

4.       Line 83: Edit: utilized

5.       Line 83-92: Decide if past or present tense will be used and be consistent

6.       Line 102-104: Sentence needs grammatical improvements

7.       Line 106-107: Suggested edit: “We hypothesize that perceived BS…”  

8.       Line 110: Edit: strengthens

9.       Line 115: Edit: Limits

10.   Line 120: Sentence needs grammatical improvements. Also, “gentle trap” is not a known term.

11.   Line 132: Edit: less career guidance and fewer opportunities

12.   Line 135-137: Sentence needs grammatical improvements

13.   Line 181: Edit: affective; influences

14.   Line 200: Sentence needs grammatical improvements

15.   Line 210: Sentence needs grammatical improvements

16.   Line 238: There appears to be a typo

17.   Line 253: Edit, remove “with”

18.   Line 263: Edit: bolstering

19.   Line 269: Edit: From an emotional perspective

20.   The BS acronym seems to disappear in the results section. Suggest being consistent with the use throughout

21.   Line 482: I believe the hyphen

22.   Line 512: I think you mean moderating factor, not meditating factor (career development strategies was a moderator in your model)

Author Response

Major Comments:

Comments 1.       The title could be improved by making more specific to the topic: The Impact of Benevolent Sexism on Women’s Career Growth: A Moderated Serial Mediation Model

Response 1: The title has been revised to The Impact of Benevolent Sexism on Women’s Career Growth: A Moderated Serial Mediation Model to better reflect the content and focus of the study. (Line 2)

Comments 2.       Line 33: “Rebound effect” and “double bias” are not known terms and should be defined/explained if used.

We have removed the terms “rebound effect” and “double bias” from the manuscript to enhance clarity and avoid confusion. The revised text now directly describes the challenges women face when exhibiting leadership-related traits, ensuring that the meaning is clear without the use of undefined terminology. (Line31-34)

Comments 3.       Line 54: “Traditional sexism” should be defined and contrasted with HS and BS.

We have removed the term “traditional sexism” from the manuscript. The revised text now focuses on BS and its impacts, ensuring clarity without introducing unnecessary terminology.(Line 55-56)

Comments 4.       Line 55: “Incivility” is not a known term and should be defined.

We have removed the term “Incivility” from the manuscript.The revised text now focuses on BS and its impacts, ensuring clarity without introducing unnecessary terminology.(Line 61-63)

Comments 5.       Line 113: What is meant by “intimate relationships” in this context?

To address this, we have clarified the meaning of “intimate relationships” in the revised text. Specifically, we now refer to “romantic partnerships and family responsibilities” to provide a clearer and more precise context. (Line 117-121)

Comments 6.       Line 124: What is meant by “masculine positions”?

We have clarified the meaning of “masculine positions” in the revised text. These positions are described as traditionally male-dominated roles that emphasize leadership, assertiveness, and competitiveness, contrasting with traditionally feminine positions such as teaching, nursing, or administrative roles.(Line 127-136)

Comments 7.       Line 163: It’s unclear how one’s cognitive understanding of a situation is self-esteem. Is it more accurate to say that one’s cognitive appraisal of the situation may impact self-esteem?

Response: The text has been revised to state that one’s cognitive appraisal of a situation may impact their self-esteem. This revision ensures a more precise explanation of the relationship between perception and self-worth (Line 170-177).

Comments 8.      Line 199-200: The meaning of this sentence isn’t clear to me.

Response:The sentence has been rephrased for clarity. It now states: Emotional exhaustion refers to a state of being emotionally and physically drained due to prolonged workplace demands. It is characterized by persistent fatigue, a lack of motivation or enthusiasm for work, and diminished effectiveness in task performance. (Line 202-205)

Comments 9.      Line 202: What are examples of “stable and specific individual behaviors” that would result from anxiety and self-doubt in response to BS?

Examples of such behaviors have been added, such as avoiding leadership roles, hesitating to voice opinions, or conforming to traditional gender expectations due to fear of judgment. These examples are now included in the Literature Review (Line 207-211).

Comments 10.  Line 215-215: It’s unclear how BS is a threat to resources.

We have clarified this point by explaining how BS imposes additional emotional demands on women, requiring them to expend greater self-regulatory resources and engage in emotional labor, which reinforces gender stereotypes and exacerbates emotional strain.(Line 220-229)

Comments 11.  Line 222: It’s unclear what is meant by “overextend their self-control” or “emotionally defensive

We have revised the text to explicitly define these terms. The revised sentence now reads:When female employees perceive BS, they often need to suppress their genuine emotional responses to meet workplace expectations. This prolonged suppression, referred to as “overextending self-control” depletes their emotional resources[56]. Simultaneously, efforts to protect their self-worth can lead to an “emotionally defensive” state, wherein they prioritize coping strategies over proactive engagement at work. This defensive state reduces their willingness and ability to invest effort into workplace tasks, ultimately lowering work performance and hindering career development[57].(Line 232-242)

Comments 12.   Line 243: Self-regulation needs to be defined in the manuscript

We have incorporated a definition of self-regulation in the manuscript. It now reads:
“Self-regulation can be defined as the uniquely human ability to consciously control and adjust one’s thoughts, emotions, impulses, and behaviors to align with social expectations or personal goals. ”(Line 259-263)

Comments 13.   Line 246-248: This sentence is confusing and needs grammatical improvements. Also, what is meant by visibility?

Response:
We have rewritten the sentence and defined “visibility” for clarity. The revised text reads:
“Career development strategies, such as seeking feedback and taking on high-profile projects, enhance visibility by increasing an individual’s presence and recognition within the workplace. Greater visibility leads to more opportunities for career advancement.”(Line 264-274)

Comments 14.  Line 260: I think prejudiced is being misused in this sentence. I think the authors mean to say that something feeling like there is a prejudiced towards them may cause short term impairment to self-esteem. As it reads now, it’s as if the individual has a prejudiced attitude themselves. This notion applies to the entire manuscript, as detailed in the next comment.

Response:
We have replaced “prejudiced” with more appropriate phrasing to clarify the meaning. The revised text now reads:

Specifically, perceived BS from surroundings may cause a short-term impairment to one's self-esteem, whereas individuals can utilize subjective cognition and self-nomination strategies to mitigate the damage caused by BS, and enhance self-recognition and self-validation, thus bolster their self-esteem[36].(Line 277-282)

Comments 15.  Throughout the Literature Review and Hypotheses section, I think it’s important to distinguish when the evidence presented about BS is referring to BS that is perceived by the individual woman and perpetrated by someone else compared to the effect of that individual woman’s own BS beliefs or actions.

Response:
We have revised the Literature Review and Hypotheses sections to clarify this distinction. Perceived BS (experienced from others)are mainly discussed in this study.

Comments 16.  Line 275: This sentence has hints of BS baked in (that women are in greater need for emotional support). Was that intended?

Response:
We appreciate your observation. The sentence has been rephrased to avoid reinforcing BS stereotypes. It now reads:Research found that women often use seeking guidance and mentoring and expanding personal networks[59].(Line 293-295)

Comments 17.   Line 293: What is meant by “representative” sample? What population were you aiming to represent?

In the revised text, we have replaced the term “representative sample” with a clearer description. The study aimed to recruit a diverse sample representing working women in urban settings, with participants drawn from various industries, career stages, and demographic backgrounds. This change ensures that our sampling approach and target population are clearly communicated.(Line 309-316)

Comments 18.   Line 300: What is “common method bias?”

We have added a definition of “common method bias” in the revised text. Specifically, we explained it as the measurement error that arises when both the predictor and criterion variables are collected from the same source at the same time. This clarification ensures that the purpose of the three-stage data collection process is clear.(Line 319-322)

Comments 19.   It would be helpful to specify where the study took place (China? Specific areas of China?-this is in the abstract but not manuscript), given there may be cultural differences in workplace behaviors, gender roles, industries, etc. This could also be noted as a limitation/future direction (expand to other areas of the world where gender roles differ)

Response:
We have updated the manuscript to specify that the study was conducted in mainland China across 18 companies in various regions.(Line 313-314)

Comments 20.  There should be a section in the Methods describing the statistical/data analysis plan. This information should not be found solely in the results.

Response:
We have divided the Methodology part into three sections: study design, participants and variable measurement. In the study design part we thoroughly describe the statistical analysis method and data collection process. (Line 308-325)

Comments 21.  I’m worried that the Benevolent Sexism scale used does not differentiate between one’s experiences of being the subject of BS versus their own sexist personal beliefs. The example item suggests that the scale assesses the latter, i.e., if I hold sexist beliefs, as opposed to “have other people’s sexists beliefs been thrust upon me. This is important because it changes the interpretation of the findings. For instance, the negative correlation with self-esteem would suggest that if I hold sexist beliefs, I’m more likely to have poor self-esteem versus, if I experience others having sexist beliefs, then I’m more likely to have poor self-esteem. This is a major difference.

Response:

Thank you for your thoughtful feedback and for raising this important point. Below, I address your concerns regarding the differentiation between personal endorsement of benevolent sexism (BS) and experiences of BS in the organizational context:

  1. Clarification on Scale Design and Question Framing
    In our questionnaire, we specifically framed the items with contextual qualifiers such as “In your organization” or “You feel that,” to ensure that respondents’ answers reflected their perceived experiences of BS within their organizational environment, rather than their personal sexist beliefs. For example, items included:

“In your organization, men typically take leadership roles because they are perceived as better equipped to handle difficult decisions.”

“In your organization, women tend to work fewer hours than men to focus on family responsibilities.”

“In your organization, you feel that when employees have personal issues, they are more likely to seek emotional support from female colleagues.”

This wording was intentionally designed to measure respondents’ perceptions of the BS present in their organizational context, ensuring the focus was on external experiences rather than internal beliefs.

  1. Distinction and Interrelation Between Perceived BS and Personal Endorsement
    While the questionnaire aims to measure respondents’ experiences of BS in their organizations, we acknowledge that such experiences can influence their own acceptance of BS to varying degrees. Prolonged exposure to a culture where BS is normalized may lead to an internalization of these beliefs, thereby affecting self-perceptions and career-related attitudes. Although distinct, the interrelationship between external BS and personal endorsement cannot be entirely disentangled, and we have aimed to account for this in our theoretical framework and analysis.
  2. Impact of BS on Women’s Self-Esteem
    This study focuses on how women’s perceptions of BS in their organizational context influence their self-esteem. Findings suggest that prolonged exposure to an environment where BS is prevalent—either through protective or patronizing behaviors—can undermine women’s self-confidence. For example, women in such environments may come to doubt their ability to handle challenging tasks or internalize the belief that difficult roles (e.g., sales or front-linepositions) are better suited to men, while women are relegated to less demanding roles (e.g., administrative or financial positions). Over time, these organizational norms can erode women’s competitiveness in critical roles and negatively impact their career trajectories.

Comments 22.   Line 363: Clarify what type of correlation testing was performed. (Spearman’s due to the ordinal nature of the data?)

Response:
In our study, we used Spearman's rank-order correlation to analyze the relationships between variables. This choice was made due to the ordinal nature of the data collected through the Likert-scale responses, which are non-parametric and do not meet the assumptions of normal distribution required for Pearson’s correlation.

Comments 23.  Suggest citing how SPSS was used to do this analysis and cite appropriate. I assume it was the PROCESS macro by Hayes?

Response:
We confirm that the data analysis was conducted using SPSS with the PROCESS macro (v4.1) developed by Andrew F. Hayes. PROCESS was used to test the moderated serial mediation model. In the revised text, we have added a citation to Hayes's foundational work on PROCESS and clarified its use in this study.(Line 409-410)

Comments 24.   Line 299-404: It would be helpful to have the variable names, at least in parentheses) for X, Y, M1 and M2 to make this paragraph more understandable without having to refer to the table.

Response:
To improve the readability of this paragraph, we will add specific variable names or as notes for X: benevolent sexism, Y: career growth, M1:self-esteem, M2: emotional exhaustion, in lines 307–465 to minimize the need to refer to the table.

Comments 25.  Line 420: If I’m understanding the result correctly, this paragraph could be said more concisely and clearly. Something like: The negative relationship between BS and self-esteem was only significant for those with low levels of career development strategy (insert values) and was not significant for high career development strategy levels.

Response:
We revised the description in line 428 to make it more concise and clear. The revised text will be something like:The negative impact of benevolent sexism on self-esteem was significant only for individuals with low levels of career development strategy (-0.306, 95% CI = [-0.447, -0.166]) and non-significant for those with high levels (0.060, 95% CI = [-0.084, 0.203]). These findings support H7a (Figure 2).

Comments 26.   How was career development strategy dichotomized for the analysis? Was a cutoff score used?

Response:
In our analysis, career development strategy (CDS) was treated as a continuous variable and was not dichotomized. This approach allowed us to capture the full variability in the data and conduct more nuanced analyses by avoiding the information loss associated with dichotomization.

Comments 27.   How were each of the scales scored? Sum? Average? What was the range of scores possible?

Response:
Each scale was scored using the mean of all items to reflect the overall trend. The score range depended on the number of items in the scale and the Likert scale used. We will include these details in the revised manuscript.

Comments 28.   Was there a sample size or power analysis performed? How big of a sample was needed for this complex analysis?

Response:
Thank you for raising this important point. Since the data analysis was conducted using PROCESS v4.1 by Andrew F. Hayes in SPSS. To ensure sufficient statistical power for the moderated serial mediation analysis, a power analysis was performed for multiple regression with 4 predictors. Assuming a medium effect size (f² = 0.15), a significance level of 0.05, and a desired power of 0.80, the minimum required sample size was calculated to be 85. Considering the complexity of the study design, we targeted a larger sample size of at least 300 participants. Ultimately, 410 valid responses were collected, ensuring robust and reliable results.

Comments 29.   Line 446: I would be careful about saying the enhancing career development strategies promotes career growth, since the analysis showed that in those with higher career development strategy scores, there was no significant relationship (neither positive nor negative) between BS and career growth (unless I misunderstood the results).

Response:
Thank you for this reminder. We will revise the statement to reflect the results more accurately. For example, “ Employees with lower levels of career development strategies experience more significant negative impacts, and enhancing career development strategies can help mitigate these adverse effects.”Further research is needed to verify whether enhancing career development strategies directly promotes career growth, particularly by examining the mechanisms at different strategy levels.

Comments on the Quality of English Language

Minor Comments:

Comments 1.       Line 24-26: The first sentence of the introduction requires grammatical improvements.

Response:
We have restructured the sentence to improve grammar and clarity. The updated version has been included in the revised manuscript.(Line 24-28)

Comments 2.       Line 49: Sentence is missing a word (“it ultimately upholds”)

Response:
We have added the missing word for proper sentence construction.

Comments 3.       Line 56-57: Sentence needs grammatical improvements

Response:
We have revised the sentence to address the grammatical issues. The updated version ensures better readability.

Comments 4.       Line 83: Edit: utilized

Response:
We have replaced the current wording with “utilized” to align with the suggested edit.

Comments 5.       Line 83-92: Decide if past or present tense will be used and be consistent

Response:
We have revised the text to ensure consistency by using the present tense throughout. The updated version:Line 90-100

Comments 6.       Line 102-104: Sentence needs grammatical improvements

Response:
We have revised the sentence to address the grammatical issues.

Comments 7.      Line 106-107: Suggested edit: “We hypothesize that perceived BS…”  

Response:

We have revised the sentence.
Comments 8.       Line 110: Edit: strengthens

We have replaced the existing term with “strengthens” as suggested.

Comments 9.       Line 115: Edit: Limits

Response:
The term “Limits” has been incorporated into the revised sentence as per the suggestion.

Comments 10.   Line 120: Sentence needs grammatical improvements. Also, “gentle trap” is not a known term.

Response:
We have revised the sentence to address the grammatical issues. And “gentle trap” has been removed for the manuscript.

Comments 11.   Line 132: Edit: less career guidance and fewer opportunities

Response:
The edit has been implemented as suggested. The revised sentence now uses “less career guidance and fewer opportunities” for grammatical correctness and clarity.

Comments 12.   Line 135-137: Sentence needs grammatical improvements

Response:
The sentence has been revised for grammatical accuracy and improved clarity. (Line 145-148)

Comments 13.   Line 181: Edit: affective; influences

Response:
We have replaced the existing terms with “affective” and “influences” as recommended.(Line 190)

Comments 14.   Line 200: Sentence needs grammatical improvements

Response:
The sentence has been revised for grammatical accuracy and clarity.

Comments 15.   Line 210: Sentence needs grammatical improvements

Response:
The sentence has been revised for grammatical accuracy and clarity.

Comments 16.   Line 238: There appears to be a typo

Response:
The typo has been identified and corrected in the revised manuscript.

Comments 17.   Line 253: Edit, remove “with”

Response:
The term “with” has been removed as suggested to improve the sentence's clarity and accuracy.

Comments 18.   Line 263: Edit: bolstering

Response:
We have replaced the existing term with “bolstering” as suggested for better alignment with the context.

Comments 19.   Line 269: Edit: From an emotional perspective

Response:
The phrase has been updated to “From an emotional perspective”

Comments 20.   The BS acronym seems to disappear in the results section. Suggest being consistent with the use throughout

Response:
We have replaced “benevolent sexism” with “BS” consistently throughout the results section to maintain uniformity across the manuscript.

Comments 21.   Line 482: I believe the hyphen 

Response:
Could you kindly clarify this comment? It seems unclear what specific issue with the hyphen is being referred to. We have replaced the hyphen with another express way.

Comments 22.   Line 512: I think you mean moderating factor, not meditating factor (career development strategies was a moderator in your model)

Response:
Thank you for catching this error. The term “meditating factor” has been corrected to “moderating factor” to accurately reflect the role of career development strategies in the model.

Reviewer 2 Report

Comments and Suggestions for Authors

Many thanks to the authors for the possibility of reading their work, in general, it is a very interesting article and very well worked. However, I would like to make few observations and recommendations to improve the quality of the manuscript and its academic impact:

- Although the study addresses a relevant topic, the introduction lacks a stronger rationale as to why China was chosen as the context and working women as the population of interest. It would be valuable to include statistical data or specific references that highlight the unique characteristics of the work environment in China and how these influence the study phenomenon.

- The theoretical framework is extensive, but some sections seem disjointed or redundant. I recommend better integrating the theories and concepts so that they more clearly guide the objectives of the study. In addition, it would be useful to explicitly connect the hypotheses to the key elements of the theoretical framework to strengthen the cohesion between sections.

- Although the CAPS (Cognitive-Affective Personality System) model is mentioned theoretically, the discussion lacks a direct and detailed relationship to the empirical analyses presented. I recommend including a more in-depth discussion of how the findings align with the postulates of this model, which would strengthen the theoretical contribution of the manuscript.

- Although it is mentioned that the instruments were translated and validated, details on the psychometric properties (e.g., reliability and validity) in the Chinese cultural context are lacking. This is crucial to ensure the validity of the measures in the study and for readers to assess the robustness of the data.

- Convenience selection of the sample may introduce biases that limit the generalizability of the findings. I recommend including a more detailed discussion of the possible implications of this bias and suggesting alternative approaches for future studies, such as implementing probability or stratified sampling.

- Although the results are clear, it would be pertinent to include a more explicit discussion of the practical implications, especially for organizational leaders and policy makers. For example, how can the findings guide interventions to mitigate benevolent sexism in the workplace?

- I identified some minor typographical and grammatical errors that could be corrected to improve the presentation of the manuscript. I recommend a thorough revision of the writing to ensure consistency and clarity.

- Tables are useful, but the manuscript would benefit from a visual figure that summarizes key relationships between variables (e.g., simplified conceptual model). This would help readers quickly and effectively understand the main findings.

- The discussion could be enriched by making clearer connections between the results obtained and the proposed theoretical framework. For example, how do the results explain the hypotheses put forward from the CAPS model? This would strengthen the theoretical impact of the manuscript.

Author Response

Comments 1:  Although the study addresses a relevant topic, the introduction lacks a stronger rationale as to why China was chosen as the context and working women as the population of interest. It would be valuable to include statistical data or specific references that highlight the unique characteristics of the work environment in China and how these influence the study phenomenon.

Reply:

We have revised the introduction to address the rationale for selecting China as the study context and the population of working women. Specifically, we added a paragraph discussing how Confucian traditions deeply shape societal perceptions of gender roles in China, emphasizing the historical and cultural norms that position women in subordinate roles. These norms not only influence individual perceptions but also organizational practices, contributing to gender stereotypes and the persistence of benevolent sexism in the workplace. This cultural backdrop provides a unique lens to examine how benevolent sexism impacts women's career growth in a rapidly modernizing society.

Additionally, we incorporated relevant statistics to strengthen the rationale. For instance, as of 2024, women held 33% of senior management roles in Chinese enterprises. Moreover, in 2023, women’s average monthly salary was approximately 87.4% of men’s earnings, highlighting the persistent wage gap and structural challenges faced by women in the workplace. These figures underscore the relevance of examining benevolent sexism within the Chinese socio-cultural and organizational context.

We hope these revisions adequately address your concerns and enhance the coherence and relevance of the introduction.

Comments 2:  The theoretical framework is extensive, but some sections seem disjointed or redundant. I recommend better integrating the theories and concepts so that they more clearly guide the objectives of the study. In addition, it would be useful to explicitly connect the hypotheses to the key elements of the theoretical framework to strengthen the cohesion between sections.

Reply:

We appreciate your valuable feedback, which has greatly helped us improve the clarity and coherence of our manuscript. Below is our response to your specific suggestions:

  1. We have revised the theoretical framework section to ensure a seamless integration of key theories and concepts. Specifically, we have linked the Cognitive-Affective Personality System (CAPS)theory more explicitly to our research objectives and hypotheses. For example, we now clearly explain how BS impacts self-esteem and emotional exhaustion within the CAPS framework.
  2. We have restructured the hypotheses section to provide explicit links between each hypothesis and the theoretical framework. For instance, Hypothesis 1-6now includes a detailed explanation of how CAPS theory underpins the mediating role of self-esteem.
  3. Redundant content in the theoretical framework section has been removed, and overlapping discussions about BS have been consolidated. This has reduced unnecessary repetition and improved readability.

Comments 3. Although the CAPS (Cognitive-Affective Personality System) model is mentioned theoretically, the discussion lacks a direct and detailed relationship to the empirical analyses presented. I recommend including a more in-depth discussion of how the findings align with the postulates of this model, which would strengthen the theoretical contribution of the manuscript.

Reply:

Your insights have greatly enhanced the theoretical clarity of the manuscript. Below is our detailed response:

  1. Incorporating CAPS Theory into Findings:We have expanded the discussion section to explicitly link the findings to the core postulates of the CAPS theory. Specifically, we now detail how benevolent sexism (BS) activates cognitive-affective units (CAUs), such as self-esteem and emotional exhaustion, which subsequently impact career growth.

Evidence: In the results section, we have added an explanation of how the mediation and moderation effects align with CAPS theory’s emphasis on situational activation and individual regulation.

  1. Detailed Discussion on Theoretical Alignment:We added the content in the discussion section to highlighted the empirical validation of CAPS theory through the observed interactions between CAUs and career outcomes. This includes a direct explanation of how career development strategies (CDS) act as a boundary condition, consistent with the CAPS framework. Key findings, such as the buffering effect of CDS on BS’s negative influence, are explicitly tied to the theoretical mechanism of CAPS.

We believe these revisions have significantly enhanced the manuscript's theoretical contribution

Comments 4:  Although it is mentioned that the instruments were translated and validated, details on the psychometric properties (e.g., reliability and validity) in the Chinese cultural context are lacking. This is crucial to ensure the validity of the measures in the study and for readers to assess the robustness of the data.

Reply:

We have expanded the methodology section to provide the following details:

  1. Translation and Validation Process
    All instruments were translated into Chinese using the back-translation method to ensure both linguistic and conceptual equivalence. A bilingual expert panel reviewed the translations to confirm cultural appropriateness and semantic clarity.
  2. Psychometric Properties
    To assess the reliability and validity of the instruments in the Chinese cultural context, we conducted the following analyses:
    • Reliability: Cronbach’s alpha coefficients were calculated for each scale. The Benevolent Sexism scale showed high internal consistency (α = 0.94), as did the Career Growth scale (α = 0.93). The Self-Esteem scale (α = 0.88) and the Emotional Exhaustion scale (α = 0.76) also demonstrated acceptable reliability.
    • Validity: A confirmatory factor analysis (CFA) was performed to assess the construct validity of the scales. The hypothesized five-factor model (Benevolent Sexism, Self-Esteem, Emotional Exhaustion, Career Growth, Career Development Strategies) exhibited good fit indices (χ² = 2713.51, df = 1700, TLI = 0.90, CFI = 0.91, RMSEA = 0.04, SRMR = 0.05). These results indicate that the scales are valid for use in the Chinese cultural context.

Comments 5:  Convenience selection of the sample may introduce biases that limit the generalizability of the findings. I recommend including a more detailed discussion of the possible implications of this bias and suggesting alternative approaches for future studies, such as implementing probability or stratified sampling.

Reply:

We acknowledge that this approach may introduce selection biases, thereby limiting the generalizability of our findings. In response, we have added a more detailed discussion in the Limitations and Future Research section of the manuscript to address these concerns.

The potential biases introduced by this sampling method may affect the external validity of the study, particularly in generalizing the results to women in less accessible industries or those with different socio-economic backgrounds. We have explicitly noted this limitation in the revised manuscript and emphasized the need for cautious interpretation of the findings.

To enhance the representativeness of future research, we recommend employing probability sampling methods, such as stratified or cluster sampling, to ensure that the sample more accurately reflects the diversity of working women in China. Stratified sampling, in particular, could be used to capture women across different industries, educational levels, and geographic regions, thereby reducing sampling bias and improving generalizability.

Comments 6:  Although the results are clear, it would be pertinent to include a more explicit discussion of the practical implications, especially for organizational leaders and policy makers. For example, how can the findings guide interventions to mitigate benevolent sexism in the workplace?

Reply:

We have revised the practical implications section to include a more explicit discussion on how the findings can guide interventions to mitigate benevolent sexism in the workplace.(Line 525-544)

Comments 7:  I identified some minor typographical and grammatical errors that could be corrected to improve the presentation of the manuscript. I recommend a thorough revision of the writing to ensure consistency and clarity.

Reply:

We have conducted a thorough revision of the entire text to address these issues. This includes corrections to grammatical errors, adjustments to ensure consistency in tense usage, and improvements to the clarity of the manuscript.

8:  Tables are useful, but the manuscript would benefit from a visual figure that summarizes key relationships between variables (e.g., simplified conceptual model). This would help readers quickly and effectively understand the main findings.

Reply:

In our current manuscript, Figure 1 already serves as a simplified conceptual model designed to highlight the key relationships between benevolent sexism, self-esteem, emotional exhaustion, career growth, and career development strategies. To address your concern, we have revised the figure’s title to “the hypothesized moderated serial mediation model” and included additional explanatory text in the manuscript to guide readers through the relationships depicted in the figure. We hope this clarification addresses your concern.

Figure 1. Proposed moderated serial mediation model.

Comments 9:  The discussion could be enriched by making clearer connections between the results obtained and the proposed theoretical framework. For example, how do the results explain the hypotheses put forward from the CAPS model? This would strengthen the theoretical impact of the manuscript.

Reply:

We have revised the discussion section to explicitly connect our findings with the CAPS theoretical framework. Specifically:

  1. The discussion now includes explicit references to how each hypothesis (H1 to H7) is supported by the results and how these findings extend the CAPS model's applicability to workplace gender dynamics.
  2. We clarified the mediating roles of self-esteem and emotional exhaustion as mechanisms through which CAPS theory explains the influence of benevolent sexism on career growth.
  3. Additionally, we have emphasized how the moderating role of career development strategies illustrates the CAPS model’s assertion about self-regulation’s critical role in behavior and motivation.

Round 2

Reviewer 1 Report

Comments and Suggestions for Authors

It appears that not all of the reivisions noted in the authors' response to previous comments made it into the version I am reading:

1. The responses to the previous comments indicate that the methods have been revised, but the version I'm reading does not have the statistical analysis plan in the methods. Rather, these details still appear in the Results section. 

2. The responses to the previous comments indicate that the manuscript specifies the study was conducted across 18 companies in mainland China, but that does not appear in the manuscript.

3. The responses to previous comments say "To improve the readability of this paragraph, we will add specific variable names or as notes for X: benevolent sexism, Y: career growth, M1:self-esteem, M2: emotional exhaustion, in lines 307–465 to minimize the need to refer to the table." I do not see those changes in the text.

4. If the career development strategy item was not dichotomized, what does "low levels" mean? How can this be interpreted? Similarly, how was Figure 2 constructed if there aren't "low" and "high" groups?

5. The power analysis should be included in the methods

Minor Comments:

1. Line 206-207: This is an incomplete sentence and can be combined with the previous sentence ("specific individual behaviors, such as avoiding leadership roles..."

2. CAU can be abbreviated in the discussion. The CDS acronym doesn't need to be redefined in the discussion as it is already defined above

Author Response

Comments 1:The responses to the previous comments indicate that the methods have been revised, but the version I'm reading does not have the statistical analysis plan in the methods. Rather, these details still appear in the Results section. 

Response 1:Thank you for pointing out the absence of a clear statistical analysis plan in the Methodology section. To address this, we have revised the section to include a concise description of the statistical methods used in this study. Specifically, we added details regarding the use of SPSS version 24.0 and the PROCESS macro (v4.1) for descriptive statistics, confirmatory factor analysis, and bootstrapped mediation and moderation analyses. This update ensures that the statistical approach is explicitly stated in the Study Design subsection, providing clarity on how the hypotheses were tested and the relationships among variables were assessed. (Line 321-324)

Comments 2:The responses to the previous comments indicate that the manuscript specifies the study was conducted across 18 companies in mainland China, but that does not appear in the manuscript.

Response 2:In the revised manuscript, we have clarified that the study was conducted across 18 companies in mainland China. This information now appears in Section 3.1 (Study Design) to ensure the geographic scope is explicitly stated. (Line 309-317)

Comments 3:The responses to previous comments say "To improve the readability of this paragraph, we will add specific variable names or as notes for X: benevolent sexism, Y: career growth, M1:self-esteem, M2: emotional exhaustion, in lines 307–465 to minimize the need to refer to the table." I do not see those changes in the text.

Response 3:Thank you for your comment. We would like to clarify that due to space limitations in Table 1, we used abbreviations for the variable names. However, in Tables 2, 3, and 4, we have used the full variable names to enhance readability and avoid requiring readers to refer back to the model for clarification. Additionally, we have incorporated the full variable names into the Results section's text descriptions. Specifically: In Section 4.2 (Descriptive Statistics and Correlations), Section 4.3 (Hypothesis Testing) and Section 4.4 (Moderation Effect Testing), we have explicitly added the full names of the variables corresponding to X, M1, M2, and Y (e.g., X: benevolent sexism, M1: self-esteem, M2: emotional exhaustion, Y: career growth) to improve clarity.

Comments 4:If the career development strategy item was not dichotomized, what does "low levels" mean? How can this be interpreted? Similarly, how was Figure 2 constructed if there aren't "low" and "high" groups?

Response 4:Thank you for your insightful comment.

In our analysis, career development strategy (CDS) was treated as a continuous variable and was not dichotomized. The term "low levels" refers to values of CDS that are one standard deviation below the mean, while "high levels" correspond to one standard deviation above the mean. These values were used in the moderation analysis to illustrate the conditional effects of benevolent sexism (BS) on self-esteem (M1) at different levels of CDS.

Regarding Figure 2, it was constructed using the regression lines derived from the conditional effects table, based on CDS values at the mean and ±1 standard deviation. This approach effectively visualizes the interaction effects while preserving CDS as a continuous variable throughout the analysis. Specifically, before presenting Figure 2 in Section 4.4, we added the explanation: "To illustrate the interaction effects, regression lines were plotted for career development strategy values at the mean and ±1 standard deviation (Figure 2), maintaining it as a continuous variable." (Line 458-460)

We hope this explanation clarifies how "low levels" were interpreted and how Figure 2 was generated.

Comments 5:The power analysis should be included in the methods

Response 5:We have now added a detailed description of the power analysis in Section 3.1 (Study Design). This includes information on the parameters used (e.g., medium effect size, significance level of 0.05, and a statistical power of 0.80) and the justification for the sample size of 410 participants, which exceeds the minimum requirement of 85. These additions ensure clarity and highlight the robustness of the study design. (Line 330-337)

Minor Comments:

Comments 1:Line 206-207: This is an incomplete sentence and can be combined with the previous sentence ("specific individual behaviors, such as avoiding leadership roles..."

Response 1:We have revised the text to ensure clarity and fluency. The updated sentence now reads: "Perceived BS can undermine women's sense of competence and trigger stress responses [49]. This can lead to anxiety and self-doubt, which may transform into stable and specific individual behaviors, such as avoiding leadership roles, hesitating to voice opinions, or conforming to traditional gender expectations due to fear of judgment [16]."

Comments 2:CAU can be abbreviated in the discussion. The CDS acronym doesn't need to be redefined in the discussion as it is already defined above

Response 2:In the discussion section, we have abbreviated CAU and omitted the redefinition of CDS for conciseness.

Reviewer 2 Report

Comments and Suggestions for Authors

Many thanks to the authors for the corrections, it is noticeable how the text has been academically strengthened. However, I would like to focus on more general and formal aspects.

1. The text is dense, especially in the introduction and literature review. Consider breaking up long paragraphs to facilitate reading and ensure that each paragraph addresses a single main point.

2. Although acronyms such are defined, they appear frequently in the text. For more general audiences, consider using the full terms occasionally to maintain clarity.

3. Check for consistency in the use of italics and capitalized terms.

Thank you very much for the opportunity to make this revision.

Author Response

Comments 1:The text is dense, especially in the introduction and literature review. Consider breaking up long paragraphs to facilitate reading and ensure that each paragraph addresses a single main point.

Response 1:

We agree that breaking up the longer paragraphs would enhance readability. To address this, we have revisited the introduction and literature review sections and restructured them into smaller, more focused paragraphs. For example:

The first paragraph introduces the background of gender inequality and the “glass ceiling” phenomenon in China.

The second highlights the socio-cultural influences of Confucian traditions on gender roles.

The third discusses workplace stereotypes and their impacts on women’s career progression.

Subsequent paragraphs delve into Ambivalent Sexism Theory (AST), the study’s theoretical framework, and its contributions.

Each paragraph now centers on a single key point, which we believe makes the text easier to follow.

Comments 2:Although acronyms such are defined, they appear frequently in the text. For more general audiences, consider using the full terms occasionally to maintain clarity.

Response 2:

Thank you for your suggestion. We have reviewed the text and replaced some occurrences of acronyms with their full terms, especially in key sections such as the introduction and discussion, to ensure clarity for a broader audience. This change balances the use of acronyms and full terms for better readability.

Comments 3:Check for consistency in the use of italics and capitalized terms.

Response 3:

We have thoroughly reviewed the document and addressed the issues regarding italics and capitalization as follows:

Italics: All subheadings have been italicized to improve readability and highlight their importance.

Capitalization: Key terms such as Benevolent Sexism (BS), Self-Esteem (SE), Emotional Exhaustion (EE), and Career Development Strategies (CDS) have been capitalized uniformly throughout the text, especially when introduced or emphasized.

Terms like "self-esteem" and "emotional exhaustion" have been standardized to use sentence case throughout the manuscript, except in headings or titles where title case is applied.

These adjustments ensure the document adheres to a consistent style, improving readability and professionalism. We appreciate your insightful feedback and the opportunity to refine the manuscript further.

Thank you very much for the opportunity to make this revision.

Round 3

Reviewer 1 Report

Comments and Suggestions for Authors

I thank the authors for adding the statistical plan to the methods. The methods state that the PROCESS macro and SPSS were used, but the results describe using Mplus for the CFA. Please address this discrepency.

Author Response

Comment 1:

The methods state that the PROCESS macro and SPSS were used, but the results describe using Mplus for the CFA. Please address this discrepency.

Response 1:

Thank you for your comments and suggestions. We have carefully addressed the discrepancy you pointed out regarding the tools used for statistical analysis.

In the revised manuscript(Line 345- 349), we have clarified that SPSS version 24.0 and the PROCESS macro (v4.1) by Hayes (2022) were utilized for descriptive statistics and hypothesis testing, including bootstrapped mediation and moderation analyses. Additionally, Mplus version 8.3 was used to conduct confirmatory factor analysis (CFA) to evaluate the measurement model and validate construct validity. This clarification ensures consistency between the methods and results sections.

Thank you for bringing this to our attention.

We hope the revised version addresses your concerns.

Best regards.